

# Subglacial topography, ice thickness, and bathymetry of Kongsfjorden, northwestern Svalbard

Katrin Lindbäck[1], Jack Kohler[1], Rickard Pettersson[2], Christopher Nuth[3], Kirsty Langley[4], Alexandra Messerli[1], Dorothée Vallot[2], Kenichi Matsuoka[1], Ola Brandt[5]

[1] Norwegian Polar Insitute, Framsentret, Postboks 6606, Langnes, 9296 Tromsø, Norway
[2] Department of Earth Sciences, Air, Water, and Landscape Science, Uppsala University, Villavägen 16, 752 36 Uppsala, Sweden
[3] University of Oslo, Postboks 1047 Blindern, 0316 Oslo, Norway
[4] Asiaq Greenland Survey, Postboks 1003, 3900 Nuuk, Greenland
[5] Norwegian Coastal Administration, Kystveien 30, 4841 Arendal, Norway

*Correspondence to:* Katrin Lindbäck (katrin.lindback@npolar.no)

**Abstract.** Svalbard tidewater glaciers are retreating, which will affect fjord circulation and ecosystems when glacier fronts become land-terminating. We present high-resolution (150 m) digital elevation models of subglacial topography and ice thickness of five tidewater glaciers in Kongsfjorden (1100 km$^2$), northwestern Spitsbergen, based on airborne and ground-based surveys. Three of the glaciers have the potential to retreat by ~10 km before they become land-terminating. The compiled data set covers one of the most studied regions in Svalbard and will be valuable for future studies of glacier dynamics, geology, hydrology and fjord circulation. The data set is freely available at Norwegian Polar Data Centre (doi: 10.21334/npolar.2017.702ca4a7).

## 1. Introduction

Ocean waters around Svalbard are warming, which in combination with the overall atmospheric warming has made Svalbard's tidewater glaciers particularly vulnerable to climate change (Nuth et al., 2013). Air temperatures have increased steadily over the last four decades, similar to the rest of the Arctic (Overland et al., 2004). Summer temperatures have the strongest influence on Svalbard glacier mass balance (van Pelt et al., 2012), and the recent summer warming has led to increasing rates of mass loss (Kohler et al., 2007). The current overall mass balance for Svalbard glaciers is negative (Moholdt et al., 2010; Nuth et al., 2010; Wouters et al., 2008), with tidewater glaciers having the greatest retreat rates overall (Nuth et al., 2013). With further warming in the Arctic, we can expect continuing retreat of Svalbard glaciers, and concomitant declines in the number of tidewater calving glaciers and total length of calving fronts around Svalbard.

More than half of Svalbard's total land area of ~60 000 km is covered by glaciers (König et al., 2014). There are over 1100 glaciers larger than 1 km$^2$ on Svalbard, and of these, 163 (15%) are tidewater glaciers. In terms of area, more than 60% of all glacier fronts terminate at sea, and the total length of calving ice-cliffs around Svalbard is estimated to be ~860 km (Błaszczyk et al., 2009). A significant portion of the meltwater in Svalbard is delivered to the ocean at calving glacier fronts.



To properly model future scenarios of fjord circulation and glacier retreat, knowledge on the subglacial topography and bathymetry under the retreating glaciers is vital. Here, we present high-resolution (150 m) digital elevation models (DEMs) of the subglacial topography and ice thickness of five tidewater glaciers in Kongsfjorden, northwestern Spitsbergen, near Ny-Ålesund (78.9° N, 12.4° E).

## 1.1 Study area

Kongsfjorden is the southern branch of the Kongsfjorden–Krossfjorden system that merges towards the open sea, in a large submarine trough, Kongsfjordrenna, which channelled a fast-flowing ice stream during the last glacial maximum (Ingólfsson and Landvik, 2013; Ottesen et al., 2005). Kongsfjorden is ~20 km long and between 4 and 10 km wide, covers an area of ~200 km$^2$ and a water volume of ~30 km$^3$ (Ito and Kudoh, 1997). Maximum depth in the outer part is 350 m and 100 m in the inner fjord. The mouth of the fjord lacks a well-defined sill and is therefore well interconnected with neighboring water masses on the West Spitsbergen Shelf, including Atlantic Water (Svendsen et al., 2002). Five tidewater glaciers terminate in Kongsfjorden (Fig. 1): Blomstrandbreen, Conwaybreen, Kongsbreen (with a north and south branch around Ossian Sarsfjellet), Kronebreen, and Kongsvegen. Kronebreen is among the fastest-flowing glaciers in Svalbard, with speeds up to ~3 m d$^{-1}$ (Schellenberger et al., 2015). Upglacier from Kongsbreen and Kronebreen are two large icefields Holtedahlfonna (named Dovrebreen in the upper part) and Isachsenfonna.

## 2. Data and methods

The main tool for mapping glacier beds is ice-penetrating radar (Dowdeswell and Evans, 2004). Our analysis builds on extensive radar campaigns conducted in the area from 2004 to 2016 (Fig. 1). Earlier campaigns (1988 to 2005) have covered the upper parts of the glaciers, but the airborne radar failed to detect the bed in the lower reaches. This was caused by too high radar frequency (dictated by limitations on antenna size on an airplane), too high travel speed with respect to the data acquisition rate, as well as radar clutter from the rough surface, crevasses and water within the glacier (Hagen & Sætrang, 1991). In recent years, radar surveys have successfully detected the bed in the lower parts of the glaciers using a lower frequency set-up mounted on a helicopter frame (Fig. 2 and 3). In the following sections, we describe the methods used to collect, process and interpolate the radar data sets into the final products of gridded subglacial elevation and ice thickness. Surveys of crevassed glaciers from helicopters is less common (e.g. Blindow et al., 2012; Kennett et al., 1993; Langhammer et al., 2017; Rutishauser et al., 2016) than surveys from fixed-wing airplanes (e.g. Bamber et al., 2013; Fretwell et al., 2013). Therefore, we describe the setup in detail.



### 2.1 Radar systems and uncertainties

### 2.1.1 Radar data collected after 2014

During early spring (April to May) in 2014, 2015 and 2016 we collected ~1300 km of common-offset radar profiles with an impulse radar system that was either suspended under a helicopter for crevassed areas (Fig. 2) or towed behind a snowmobile.

The system is based on a radar developed for surveying of ice thickness on the Greenland ice sheet (Lindbäck et al., 2014). The radar system consisted of resistively loaded half-wavelength dipole antennas of 10 MHz centre frequency. We used an impulse transmitter with an average output power of 35 W and a pulse repetition frequency of 1 kHz. The trace acquisition was triggered by the direct wave pulse between transmitter and receiver. The 14-bit A/D converter sampled two channels at 125 MHz sampling frequency, with different sensitivity ranges and one channel was attenuated with 20 dB to record both the

surface and the bed return.

Using the helicopter-based system we surveyed the glaciers at a nominal speed of ~40 km h$^{-1}$, along tracks separated by ~0.5 to 1 km. By stacking 125 traces, a mean trace spacing of 4 m was achieved. We mounted the system on a 3 x 3 m wooden frame, with extended wood arms and plastic rods for the antenna (Fig. 3). The frame was suspended 20 m below the helicopter. The radar was controlled by wireless connection to a PC inside the helicopter. We used the ground-based system to survey the

15 snowmobile accessible Kongsvegen and Holtedahlfonna. The system was mounted on two sleds and towed behind the snowmobile at a speed of ~20 km h$^{-1}$, with an antenna separation of 30 m and resulting mean trace spacing of 2 m.

We positioned the traces by using data from a code-phase Global Positioning System (GPS) receiver in 2014 and 2015 and a carrier-phase dual-frequency GPS receiver in 2016, mounted 1.5 m from the common midpoint along the travelled trajectory on the helicopter frame and 15 m from the common midpoint on the snowmobile. For the dual-frequency receiver we processed

the data kinematically using the Canadian Spatial Reference System precise point positioning service (Natural Resources Canada, 2017).

We applied several corrections and filters to the radar data: (1) a Butterworth bandpass filter, with cut-off frequencies of 2 and 50 MHz, to remove unwanted frequency components in the data, (2) normal move-out correction to correct for antenna separation, (3) rubber-band correction to get the data to uniform trace spacing, and (4) two-dimensional frequency wave-

25 number migration (Stolt, 1978) to collapse hyperbolic reflectors back to their original positions in the profile direction. On the high gain channel, we applied a spherical and exponential (SEC) gain to reinforce bed returns. The surface and bed returns were digitized semi-automatically with a cross-correlation picker (Irving et al., 2007) at the first break of the bed reflection. We calculated ice thickness from the picked travel times of the bed return using a constant radio-wave velocity of 168 m μs$^{-1}$ for ice. For the airborne profiles, we removed the travel times to the surface return using a constant radio-wave velocity of 300

30 m μs$^{-1}$ for air. We converted the GRS80 ellipsoidal heights to heights above sea level with a geoid model developed by Norwegian Polar Institute, where the average geoid height is ~35 ± 0.5 m in the study area relative to the ellipsoid. Figure 4 shows examples of processed radar images.





### 2.1.2 Radar data collected prior to 2014

In addition to the data collected in this study, we used two additional data sets of unpublished radar data collected earlier by the Norwegian Polar Institute on: (1) Dovrebreen in 2004 and 2005, and (2) Kronebreen and Holtedahlfonna in 2009 and 2010. We did not use additional data sets collected in Kongsvegen in 1988 (ice thickness; Hagen & Sætrang 1991) and 1995

(subglacial elevation and ice thickness; Melvold & Hagen 1998), because these data sets had large differences in subglacial elevation. This is probably because the data were collected over 20 years ago and possibly significant changes in glacier surface and subglacial sediment may hinder accurate estimates of subglacial elevations from these old ice-thickness data. Here follows a short summary of the two included data sets:

*The Dovrebreen campaign:* The data set consists of 33 km of radar profiles collected on the upper parts of Dovrebreen, during

an ice coring campaign (Beaudon et al., 2013; Sjögren et al., 2007). The subglacial elevation and ice thickness were measured in April 2004 and 2005, with a 10 MHz center frequency impulse radar. A single channel impulse radar based on a Narod transmitter (Narod and Clarke, 1994) and a 12-bit A/D converter in the receiver were used with restively loaded dipoles as antennas. The radar was operated at both 100 MHz sampling frequency and at 200, 300 and 500 MHz sampling frequency using repetitive sampling. The repetitive sampling gave a non-perfectly uniform sampling frequency in the whole scan and the

data had therefore been resampled to equal time base between the samples with linear interpolation. An antenna separation of 20 m was used. The antennas were configured in an end-fire mode with the transmitter in the back and the receiver approximately 25 m behind a snowmobile. The positioning of the profiles were made with a code-phase GPS receiver attached to the radar receiver and the position was recorded each second.

*The Kronebreen and Holtedahlfonna campaign:* The data consists of 4800 km of radar profiles collected in 2009 to 2010,

where both helicopter and snowmobiles were used. Data were collected with an impulse dipole radar comprising a Kentech pulser (±2000 V), 10 MHz resistively loaded wire dipole antennas, and a 12-bit A/D converter. The helicopter and ground-based system was similar to the one previously described (see Section 2.1.1). The A/D converter sampled with two channels at 50 MHz sampling frequency. Five traces were stacked in flight, and further stacking was done during post processing. Positioning was made with a code-phase GPS receiver attached to the radar receiver.

**2.1.3 Radar system errors and uncertainty**

We used standard analytical error propagation methods (Lapazaran et al., 2016; Taylor, 1996) to calculate the error in subglacial elevation for each data point:

$$\varepsilon_{bed\ data} = \sqrt{\varepsilon_{radar}^2 + \varepsilon_{xy}^2} \tag{1}$$

where $\varepsilon_{radar}$ was the error in the radar acquisition and $\varepsilon_{xy}$ was the positioning error. The error in radar acquisition was

calculated by:

$$\varepsilon_{radar} = \frac{1}{2}\sqrt{v^2 \cdot \varepsilon_t^2 + t^2 \cdot \varepsilon_v^2} \tag{2}$$





where $v$ was the radio-wave velocity used for time-to-depth conversation, $t$ was the two-way-travel time of the radio wave and $\varepsilon_t$ and $\varepsilon_v$ were the errors in $t$ and $v$ respectively. We used a constant wave-propagation speed for the ground-based and airborne surveys (168 m μs$^{-1}$). Wave velocity can vary spatially depending mostly on density. Profiles were collected in the ablation and accumulation zone with a snow and firn cover up to 20 m thick in the upper parts of Dovrebreen (Beaudon et al.,

2013). We used a typical variation of 4% of glacier ice density for the calculation of $\varepsilon_v$ (Seligman, 1936). Variations in the wave velocity can also occur because of varying ice temperature and the presence of inhomogeneities and liquid water in the ice (Drewry, 1975). These effects are expected to have a small impact on the average velocity for the whole ice column, while water content in the ice can influence the velocity in a substantial way. In most parts of the study area the ice is cold (Beaudon et al., 2013) and there are limited amounts of liquid water. We therefore neglect variations of velocity due to water content.

The upper parts of Holtedahlfonna contain a firn aquifer (Christianson et al., 2015), but it comprises a small part of the total glacierized area, and is not accounted for. For $\varepsilon_t$ we calculated the range resolution, which is the accuracy of the measurement of distance between the antenna and the bed and can be determined from the characteristics of the source pulse (i.e. bandwidth) and the digitization frequency. The range resolution for the data collected in this study was estimated at 8.5 m. We also included the vertical resolution, by taking the inverse of the radar frequency (Lapazaran et al., 2016). This results in values of $\varepsilon_{radar}$

between 8.5 (thin ice) and 30.5 m (thick ice) with a mean value of 14.3 m and standard deviation of 4.2 m (Fig. 5a). We calculated the positioning error $\varepsilon_{xy}$ at each data point depending on the bed slope angle along the profile, following the method by Lapazaran et al. (2016). We assumed a helicopter-travel speed of 40 km h$^{-1}$, snowmobile-travel speed of 20 km h$^{-1}$, and $T_{GPS} \leq T_{GPR}$ (case a' in Appendix B, Lapazaran et al. 2016). This produced values of $\varepsilon_{xy}$ between 0 (flat bed) and 76.0 m (steep bed ~90°), with a mean value of 1.4 m and standard deviation of 2.1 m (Fig. 5b). The total error in subglacial elevation

along the profiles (Eq. 2) varied between 8.5 and 78.0 m, with a mean of 14.5 m and standard deviation of 4.3 m (Fig. 5c).

To test the consistency between the data sets we also compared the crossover differences in the subglacial elevation estimates between different profiles and data sets. The data set collected in this study (2014 to 2016) had a median crossover misfit in subglacial elevation of 11.1 m with a standard deviation (σ) of 17.4 m based on 208 crossing points. The Dovrebreen campaign data set (2004 to 2005) had a median crossover misfit of 13.1 m (σ = 12.7 m) based on 85 crossing points. The Kronebreen

and Holtedahlfonna campaign data set (2009 to 2010) had a median crossover misfit of 3.9 m (σ = 13.0 m) based on 136 crossing points. As the crossover analysis within the same data set does not capture systematic errors between the different data sets, we also did a comparison between the data sets. When we ran a crossover analysis between all the data sets the median misfit was 9.2 m (σ = 15.7 m).

## 2.2 Surface and bathymetric elevation data

To obtain surface elevation for the study area including the non-glaciated area we used TanDEM-X monostatic radar images acquired on December 20, 2014 processed interferometrically using Gamma Software. The monoscopic images were first co-registered to each other and to a previous baseline DEM derived from the TanDEM-X intermediate DEM. The differential



phase was unwrapped to provide elevation differences, which were then added back to the original baseline DEM to provide elevations from December 20, 2014. These elevations were for the underlying ice and ground surface as the X-band satellite radar wave can penetrate through the winter snowpack providing a reflector from the ground and ice interface. On stable terrain surrounding the glaciers, the 2014 TanDEM-X DEM shows little bias, with a standard deviation of ~4 m compared with the

2009 aerial-photogrammetric DEM from the Norwegian Polar Insitute (NPI, 2014), which has a 5-meter resolution and a stated accuracy of ~2 to 5 m. Elevation changes between 2009 and 2014 have mostly occurred close to the margins of the tidewater glaciers (<5 km), with up to 5 m of surface lowering (pers. comm, Cesar Deschemps). The NPI (2014) DEM was also used to cover Blomstrandbreen, which is located outside of the coverage of the TanDEM-X DEM.

The bathymetric DEM was compiled by Norwegian Mapping Authority Hydrographic Service. Data was captured in 2000

with an EM 1002 multi-beam echo sounder. In 2007, 2010 and 2011 an EM 3002 multi beam echo sounder was used. They derived the 5 m grid DEM in 2014 with the software QPS Fledermaus and CARIS HIPS/CARIS BathyDataBASE.

## 2.3 Assimilation of the data sets

We combined the different data sets of subglacial, land and bathymetry elevation to a final gridded elevation DEM. The measuring interval for the radar data sets are dense along the profiles, with a data point spacing of ~5 m, compared with ~50

to 1000 m spacing between individual profiles. As this non-uniform spacing is not optimal for gridding algorithms, we subgridded data sets into a 100 m pseudo-grid to reduce the data density along individual profiles. The subgrid was produced by calculating the median values for the points that fell within the distance of half the grid cell. To prevent steps at the borders between the subglacial and proglacial DEMs we point sampled land-topography and bathymetric DEMs in a buffer zone around the glacier margins, which were added to the subgrid. We used a universal kriging algorithm (e.g. Isaaks & Srivastava

1989) for the interpolation. To calculate glacier ice thickness in 2014, we subtracted the subglacial DEM from the combined TanDEM-X DEM and NPI DEM. We did this instead of using ice thickness measurement for each radio echo sounding data set, to make the DEM more consistent over the area, as the ice thickness has changed since the first data were collected in 2004.

The subglacial DEM agrees well with a borehole study in Kronebreen in 2014 (How et al., 2017), with a measured subglacial

elevation of −93 m a.s.l., where the gridded DEM predicts a depth of 90 m. To assess the error in interpolation we cross-validated the gridded data, which is a common validation technique to see how well an interpolated model is influenced by the observed data. By removing one observation from the data set, the remaining data were used to interpolate a value for the removed observation. This process was continued for 1000 random observations in the data, where the error is the residual between the observed and the interpolated value (Isaaks and Srivastava, 1989). The standard deviation of the residuals was

estimated to be 18 m, and increases with distance from the profiles. To summarize, the total accuracy of the gridded subglacial elevation depends on: (1) the technical and theoretical capability of the radar systems; (2) positioning errors; and (3)





interpolation errors. By assessing all these potential sources of error, we estimate the maximum vertical root-mean-squared uncertainty in the final DEM to be approximately ±24 m.

## 3. Results

We present DEMs of subglacial topography, bathymetry (Fig. 6a) and ice thickness (Fig. 6b) of a 1100 km$^2$ area of Svalbard
at a 150-m spatial resolution (high-resolution images are available in Supplements). The DEMs cover the tidewater glaciers Blomstrandbreen, Conwaybreen, Kongsbreen, Kronebreen, and Kongsvegen and are merged with bathymetric and land DEMs for the non-glaciated areas. The large-scale subglacial topography of the study area is characterized by a series of troughs and highs. The minimum subglacial elevation is −180 m a.s.l., the maximum subglacial elevation is 1400 m a.s.l. and the maximum ice thickness is 730 m. The statistics for each glacier are summarized in Table 1. Kronebreen, which has a dense data coverage
(Fig. 1), we also present a 50 m spatial resolution subglacial DEM (Fig. 7).

## 4. Discussion

To properly model future scenarios of fjord circulation and glacier retreat, knowledge on the subglacial topography and bathymetry under the retreating glaciers is important. In the following sections, we shortly discuss the main troughs and highs in the subglacial topography.
Overdeepenings in Blomstrandbreen and Conwaybreen (−110 resp. −60 m a.s.l.; Fig. 6a) lie behind sills at the glacier front, which limit the extension of the fjord further upglacier; both overdeepenings will therefore likely be either filled with sediments or freshwater, as the glaciers retreat. Further south, Kongsbreen consists of two tributary outlets around the Ossian Sarsfjellet. Kongsbreen North has a deep trough beneath it with a minimum elevation of −180 m a.s.l. The continuation of the fjord (i.e., elevation beneath current sea level) extends 11 km inland from the current front to north of the nunatack Steindolpen upglacier
from Collethøgda. The fjord may possibly connect with Kronebreen in a 500 m wide embayment, with only 10 m deep waters. Kongsbreen South has a sill at its front, with a minimum elevation of 8 m a.s.l., preventing the embayment in front of the glacier connecting to Kongsbreen North. This rock outcrop is clearly visible at the glacier front in current satellite images. On the other hand, the exposed outcropping may also be small islands, with deeper elevations not imaged by radar since such topography is hard to resolve in detail with 500 m between the radar profiles and a gridded DEM resolution of 150 m.
Kronebreen, the fastest flowing glacier in the fjord (Schellenberger et al., 2015), has a trough beneath it with a minimum elevation of −130 m a.s.l. The trough continues 10 km inland, where it ends at a 350 m wide and 2 km long embayment with 20 m shallow waters. Upglacier from the embayment there is a steep sill, with a minimum elevation of 130 m a.s.l. (Fig. 6a), which can also be seen at the glacier surface, where there is a steep and heavily crevassed ice fall. Upglacier, the ice thickens again and there is a small overdeepening with a minimum subglacial elevation of −30 m a.s.l. The high-resolution subglacial

DEM of Kronebreen has so far been used in studies of basal sliding, subglacial hydrology and calving (How et al., 2017; Vallot et al., 2017, 2018).

The subglacial topography beneath Isachsenfonna consists of a 3 km wide flat valley with a minimum subglacial elevation of 40 m a.s.l. Holtedahlfonna has a 2 km wide valley with higher subglacial elevations, with a minimum elevation of 120 m a.s.l, and gradually higher elevations up on Dovrebreen (Fig. 6a). Finally, Kongsvegen flowing in from southeast towards Kronebreen, has a trough beneath it with a minimum subglacial elevation of −70 m a.s.l., which continues 9 km inland until just north of the nunatak Vorehaugen.

## 5. Data availability

The compiled data sets of ground-based and airborne radar surveys are freely available at Norwegian Polar Data Centre (doi: 10.21334/npolar.2017.702ca4a7). The data set will be updated when the quality of the data is improved or if new data sets become available.

## 6. Conclusion

Tidewater glaciers have a major influence on circulation in the water bodies in which they sit, particularly in constricted bays or fjords. In this study, we produced subglacial topography and ice thickness DEMs of five tidewater glaciers in Kongsfjorden. The subglacial topography DEM shows that the glaciers Kongsbreen, Kronebreen and Kongsvegen have the potential to retreat by ~10 km before they become land-terminating. The compiled data set covers one of the most studied regions in Svalbard and will be valuable for future studies of glacier dynamics, geology, hydrology and fjord circulation.

## Author contributions

KaL was the main responsible for collecting, processing, and analysing the data and prepared the manuscript with contributions from all co-authors. JK was the project leader and was the main responsible for fieldwork and data management. RP was the main responsible for the radar system used after 2014. KaL, JK, AM and DV collected the radar data in field (campaigns after 2014). CN provided the surface TanDEM-X data set. KiL, KM and OB provided radar data from the earlier campaigns (prior to 2014). The authors declare that they have no conflict of interest.

## Acknowledgements

This work was part of the TIGRIF (Tidewater Glacier Retreat Impact on Fjord circulation and ecosystems) project, funded by the Research Council of Norway, Oceans and Coastal Areas Programme (project 243808). Funding has also been provided by



the GLAERE project (the Polish-Norwegian Research Programme) and TW-ICE projects (Center for Ice, Climate, and Ecosystems of the Norwegian Polar Institute). Field support was given from the Swedish Society for Anthropology and Geography (SSAG), Svalbard Science Forum-SSF (RIS #6660) and the Nordic Centre of Excellence SVALI. We would also like to thank Geir Gunleiksrud and Boele Kuipers for the bathymetric DEM. CN acknowledges funding from European

Union/ERC (grant No. 320816) and ESA (project Glaciers_CCI, 4000109873/14/I-NB). The TanDEM-X DEM and IDEM were provided by the German Space Agency (DLR) satellites TerraSar-X and TanDEM-X (proposals XTI_GLAC6716 and IDEM_GLAC0435). We thank Edward King and Bryn Hubbard for reviewing the manuscript in an earlier stage.

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

| Glacier | Subgl. elevation (m a.s.l.) | Ice thickness (m) |
|---------|------------------------------|-------------------|
| *Blomstrandbreen* | Max: 1190<br>Min: −110<br>Mean: 250 | Max: 410<br>Min: 0<br>Mean: 160 |
| *Conwaybreen* | Max: 1200<br>Min: −60<br>Mean: 340 | Max: 320<br>Min: 0<br>Mean: 110 |
| *Kongsbreen* | Max: 1400<br>Min: −180<br>Mean: 250 | Max: 730<br>Min: 0<br>Mean: 250 |
| *Kronebreen* | Max: 1390<br>Min: −130<br>Mean: 160 | Max: 580<br>Min: 0<br>Mean: 280 |
| *Kongsvegen* | Max: 1010<br>Min: −70<br>Mean: 160 | Max: 450<br>Min: 0<br>Mean: 190 |
| ***Total*** | Max: 1400<br>Mean: 419<br>Min: −180 | Max: 730<br>Mean: 182<br>Min: 0 |





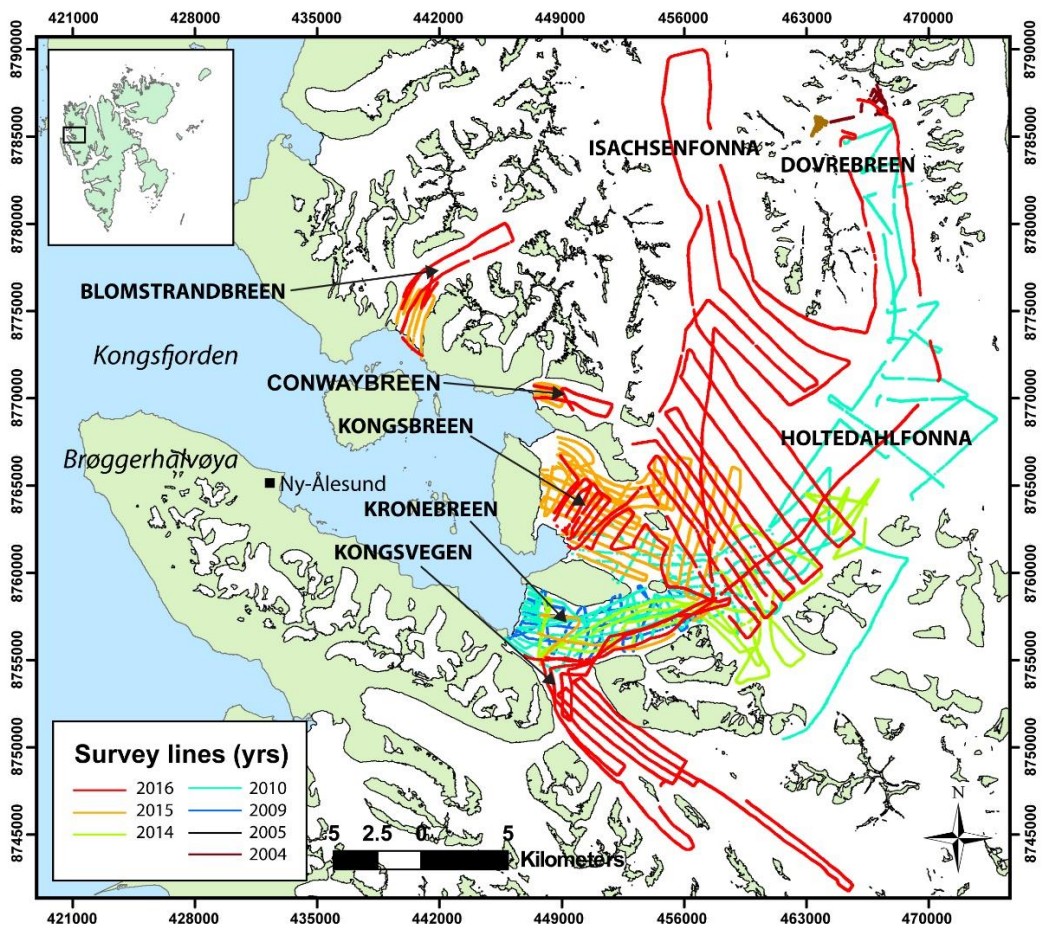

**Figure 1: Radar survey-lines collected between 2004 and 2016. Tidewater-glacier names, the peninsula Brøggerhalvøya, and the research town Ny-Ålesund are marked in the map. Small inset map of Svalbard, showing the location of Kongsfjorden. The blue areas are sea, green areas land and white glacierized (2009). The black lines indicates the location of the profiles in Figure 4. Grid projections are Universal Transverse Mercator Zone 33W.**

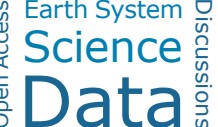

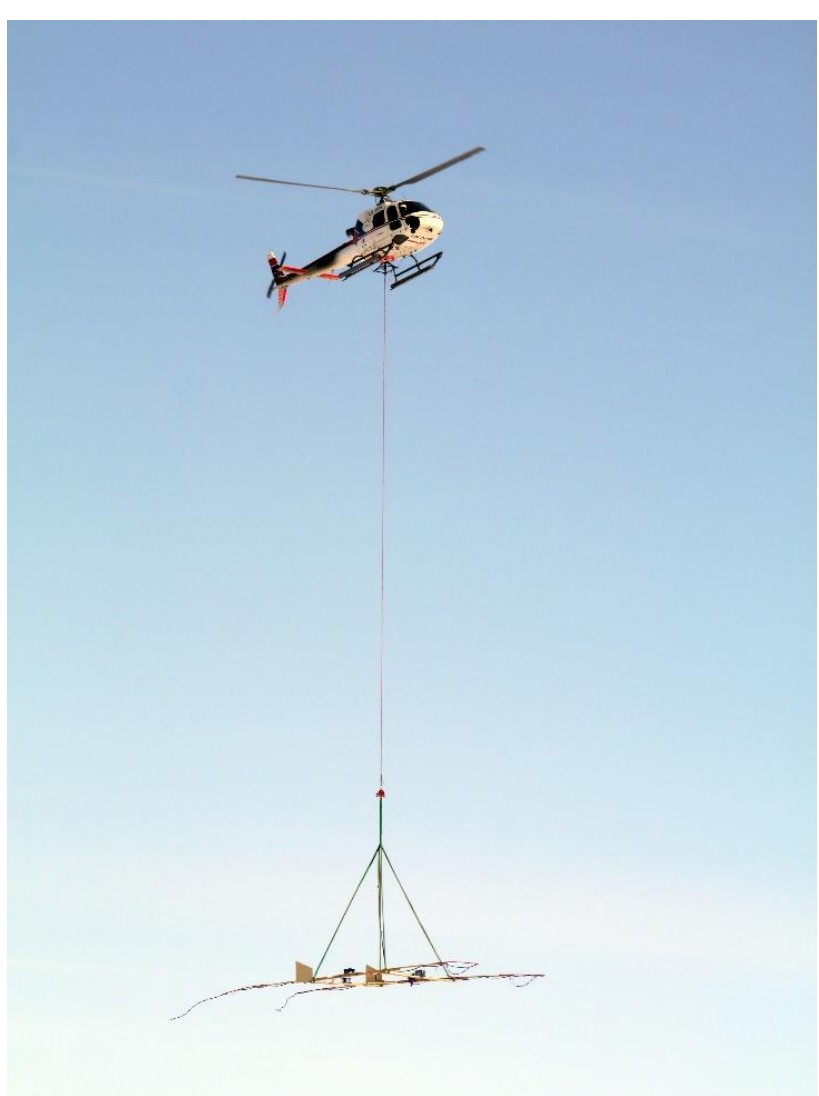

**Figure 2: Helicopter with radar frame. Photo: Nick Hulton.**




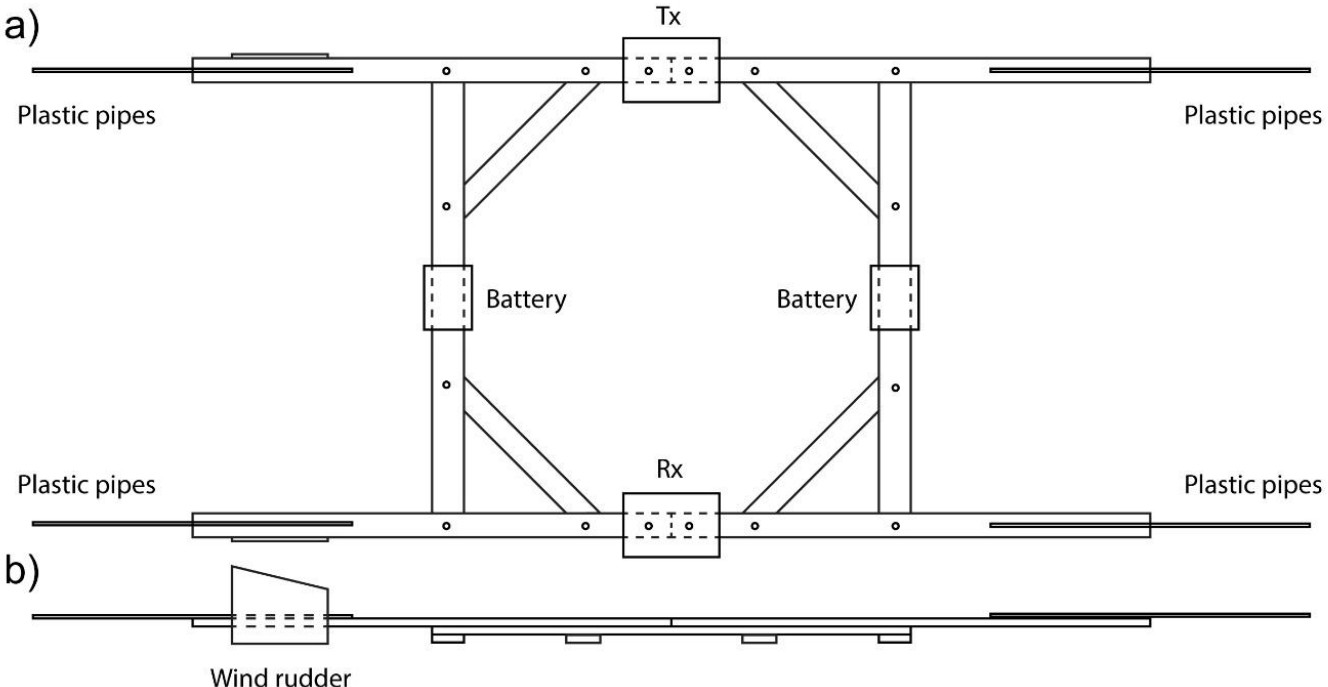

**Figure 3: Helicopter wooden frame from a) above, with transmitter slot (Tx), receiver slot (Rx), two slots for batteries and plastic pipes for holding the antennas, and b) sideways, with two fins on each side that function as wind rudders to prevent the frame from spinning.**



**Figure. 4: Examples of processed radar images collected on a) Kongsvegen by snowmobile and b) Holtedahlfonna by helicopter. The**
5   **locations of the profiles are marked in Figure 1 with black lines.**





**Figure. 5: Errors in the subglacial elevation for each data point: a) Radar errors consisting of the technical and theoretical capacity of the radar systems, b) Positioning errors, and c) The total error when combining radar and positioning errors. Grid projections are Universal Transverse Mercator Zone 33W.**

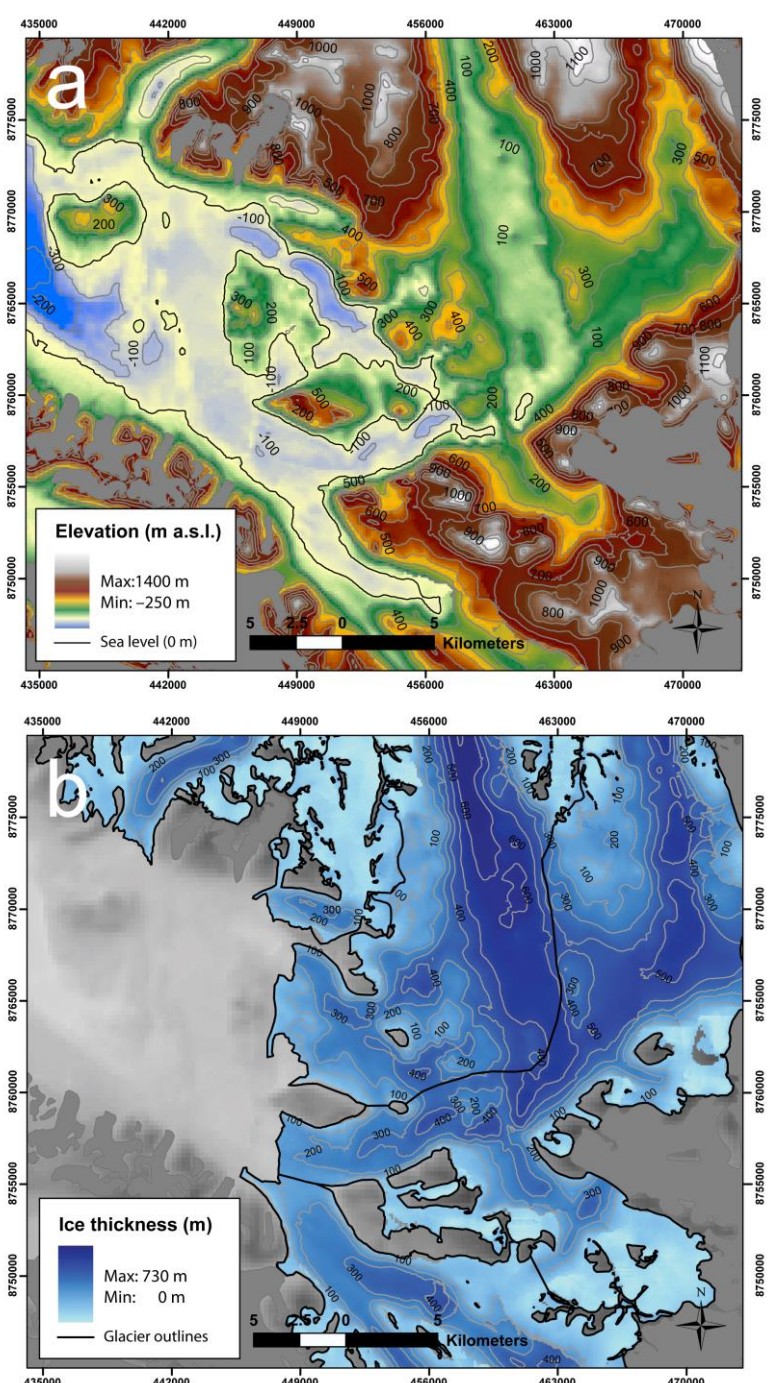

**Figure 6: a) Elevation with 100-m elevation contours. b) Ice thickness in 2014 with 100-m elevation contours. Surface elevation catchments are marked (black lines). Statistics for each glacier are specified in Table 1. Grid projections are Universal Transverse Mercator Zone 33W.**





**Figure 7: Elevation of Kronebreen at 50-m resolution with 20 m elevation contours and the location of the borehole study (black star; How et al., 2017). Grid projections are Universal Transverse Mercator Zone 33W.**

