# Peer review of "Subglacial topography, ice thickness, and bathymetry of Kongsfjorden, northwestern Svalbard"

_Earth System Science Data, 2018_

## Referee Comment (RC1) · N. Ross (Referee) · 17 May 2018

**Review of Lindbäck et al., Earth System Science Data Discussion, 2018**

This is a very good manuscript that presents ice thickness and subglacial topography/bathymetry digital elevation models (DEMs) of a series of glaciers in NW Svalbard. The quality of the data is very good, and the methods of acquisition and processing (and their description) are appropriate. Errors and uncertainties are outlined effectively. The DEMs are important for several purposes, the most important being for numerical modelling of glacier behaviour/evolution, with implications for global sea level rise.

I do have some suggestions for ways in which the manuscript can be improved before publication:

1. Abstract and conclusions are very superficial and general at present. Both need improvement so that they actually report/summarise the manuscript and provide more detail. Abstract is currently very short, so there is room to develop it.
2. The figures: (a) have multiple basic errors (e.g. lack easting northing, have bizarre distance scales etc.); (b) could be improved with some simple changes to the GIS (e.g. have discrete/classified colour scales, rather than continuous ones), and (c) are lacking the presentation of certain datasets (e.g. ice surface elevation). Detailed suggestions of how to edit/improve the figures are provided below.
3. Results section is currently perfunctory (7 lines), whilst significant parts of the discussion section simply describe the data rather than interpret or discuss it. I recommend that sections 3&4 are merged into a single section entitled something along the lines of "Description of DEM morphology and implications for future glacier and landscape dynamics in NW Svalbard"
4. The authors should consider including some profiles extracted from the DEMs as figures. This would serve 2 purposes: (a) to illustrate to those not familiar with the study area and dataset the 'morphology' of the ice thickness/bed elevation; (b) to qualitatively demonstrate the quality of the data (e.g. are there any artefacts at critical locations?). I suggest that along ice flow profiles down the centre lines of the glaciers would be useful as (a) such profiles would likely be input data for 2D ice flow models; and (b) the discussion section describes sills and overdeepenings that are not necessarily obvious from the DEMs alone (at least to someone not familiar with the datasets).

**Detailed Comments:**
**Abstract:**
- P1. L12-13: "..which will affect fjord circulation and ecosystems…." – how will retreating glaciers do this? If this is the justification for the datasets, then you need to explain how. I'd also suggest adding something about global sea level, which is the most important impact of your dataset (i.e. DEM can be used for numerical modelling of future glacier behaviour, from which future sea level can be modelled), and perhaps about insights into surging glacier behaviour.
- P1. L13-15: it is worth inserting "ice-penetrating radar" into this sentence (i.e. after "ground-based").
- P1. L15: One sentence on findings. This is not enough. I also suggest a re-phrasing to "Three of the glaciers would have to retreat by ~10 km…..". It would also be worth naming those three glaciers.
- P1. L16-17: "…will be valuable for future studies of glacier dynamics, geology, hydrology and fjord circulation". Fair enough, but how and why? The authors never discuss why the data would be valuable for many of these topics in the manuscript, so why make the statement here? Perhaps the discussion section of the manuscript could be expanded to develop this justification though?

**Introduction:**
- P1. L26-27: Perhaps add "..providing a contribution to rising global sea levels."?
- P2. L1: The glaciers could advance too (particularly if they are of surge-type), so perhaps "glacier dynamics" or "glacier oscillations" rather than "glacier retreat"?

- Do any of the glaciers in the study area surge? It might be worth stating whether this is the case or not. I believe that Kongsvegen is surge-type glacier?

**Study area:**
- P2. L9: "Maximum depth in the outer part of the fjord…."?
- I'd would have liked to have seen more information on the glaciology of these glaciers (e.g. surging, subglacial sediments, thermal regime etc.), or at least more references to published papers that describe the glaciological characteristics of these in detail (e.g. I am aware of papers by John Woodward/Tavi Murray/Adam Booth on Kongsvegen), and perhaps at least some description of the wider controls on the glacial system in the study area (e.g. temperature/precipitation/ oceanography etc.).

**Data and methods:**
- P2. L18: No need for "have"
- P2. L19-20: "…high a radar frequency…"?
- P2. L25: "are" rather than "is"?
- P2. L26: There are much better references than Bamber et al and Fretwell et al. I recommend that examples that report individual airborne surveys are referenced, rather than those that report Antarctic- and Greenland-wide DEMs.

**Radar data collected after 2014:**
- P3. L7: Can the authors provide more information on the transmitter? Later in the manuscript they refer to Kentech and Narod transmitters, but they do not describe this one. Is it a bespoke transmitter built by NPI? Please either state that it is a bespoke system, or, if it is an 'off-the-shelf' system, please give its name (e.g. Narod etc.).
- P3. L9: Break this into two sentences: "…different sensitivity ranges. One channel was attenuated by……"
- P3. L16: still 125 traces stacked (i.e. equivalent to acquisition with airborne system)?
- P3. L19: 15 m in front or behind the midpoint?
- P3. L24: "…rubber-band correction to re-sample the data to a uniform…"?
- P3. L26: "amplify" rather than "reinforce"?
- P3. L28: The velocity of the radio wave through the ice is assumed (and assumes cold ice?). Can the authors justify this assumption in anyway? I note that Woodward et al., Journal of Glaciology, 2003 reports CMP measurements on Kongsvegen that could be referenced.

**Radar data collected prior to 2014:**
- P4. L6: Delete "possibly"
- P4. L14-15: It might have just been me, but I didn't really follow this sentence. Consider rewording.
- P4. L23: how much stacking?

**Surface and bathymetric elevation data:**
- P6. L9: "The offshore bathymetric…"?
- P6. L9: "acquired" rather than "captured"?
- P6. L9-11: are there any references for this dataset (e.g. a technical report). If there are not, then perhaps a more detailed description of acquisition and processing is required within this manuscript?

**Results:**
- P7. L9: "For Kronebreen….."?
- P7. L10: Are these 50 m resolution DEMs also available via the NPI data centre website?

**Discussion:**
- P7. L12-13: Reword to: "…knowledge of the subglacial topography of retreating glaciers…"
- P7. L15-P8. L7: Lots of this material is description of the DEM morphology, so should be in a results section not a discussion section. In addition, there are multiple features (e.g. overdeepenings)/place names (e.g. Steindolpen) referred to that are not annotated on any figure. To improve readability, the authors should annotate appropriate figures.
- Much of the content describing the morphology of the DEMs would be helped by the inclusion of profiles (e.g. along ice flow, down the centre line of the glaciers) that would help to illustrate features such as overdeepenings and sills. You might also want to cite some classic literature in this section too (e.g. Holtedahl 1967 https://www.tandfonline.com/doi/abs/10.1080/04353676.1967.11879749)
- P7. L22: Perhaps include a satellite image in the manuscript, and annotate the rock outcrop? I would suggest that a satellite image of the entire study area would be useful (e.g. figure 1).
- P7. L22-24: Might have been me, but this sentence didn't seem easy to disentangle. I suggest the authors consider rewording it.

**Conclusion:**
- Like the abstract, rather generic and superficial. I suggest that the authors expand the conclusion to report on/summarise the dataset and manuscript in more detail.

**Author contributions:**
- P8. L21. "was the main responsible"? Do the authors mean "was primarily responsible"?

**Table 1:**
- Is it also worth reporting standard deviation?

**Figures (general comments)**
- All maps need 'easting (utm)' and 'northing (utm)'. Currently only figure 5 has this.
- The distance scale on all of the maps should extend upwards from 0, rather than having ranges from negative values to positive values (e.g. fig 1 extends from -5 to 5 km)
- "Grid *projection is* Universal Transverse Mercator Zone 33W"
- Maps should be annotated with features/places referred to in the text (e.g. Steindolpen nunatak)

**Figure 1**
- I could not make out the location of: (a) 2005 data; or (b) Black lines indicating location of profiles in figure 4.
- A satellite image of the study area would be useful.
- It would be useful to show which survey lines were (a) acquired by helicopter; (b) acquired by skidoo.
- Rather than stating "blue areas are sea, green areas land and white glacierized", why not add these to the legend?

**Figure 3**
- This (& fig 2) is a really useful figure. It should not be removed from the manuscript. Could figures 2 & 3 be integrated into one figure however?
- There is a little ambiguity to me about the position of the antennas. Do they run from the Tx and Rx outwards to the end of the plastic pipes, or are they just contained within the plastic pipes. Can the authors please make this clear?
- What about a scale bar?

**Figure 4**

- Where are these radagrams located? I could not see them on figure 1.
- Perhaps the authors could annotate key features in the radargrams for non-experts in ice-penetrating radar (e.g. ice surface, ice-bed interface, internal layering, hyperbolic reflections from englacial conduits etc.). Perhaps also worth annotating surface multiples?
- Figure 4a: Can the authors explain why if the data have been migrated (as stated in the methods section) there are still hyperbolic reflections within the ice column? Is this evidence for warm ice with a different velocity to that used in the migration? It looks like there might be hyperbola at the bed too (i.e. between 0-2 km).

**Figure 5**

- No units are given? I assume metres?

**Figure 6**

- Needs a box showing its location on figure 1 (the extent of figure 1 is greater than that of figure 6).
- Edit the colour scales so that they are in discrete intervals of equal spacing (e.g. 100 m intervals), so that the boundaries between the colours match the contours. At present, with a continuous colour scale it is very difficult for the reader to match the colours with actual elevations. The inclusion of some topographic/bathymetric profiles (see earlier comments) may help with this.
- The authors report a surface DEM in the manuscript. This should be displayed in this figure. They may also wish to include an ice velocity map if one exists for this region.
- 6a – it needs to be made clearer that this is the subglacial/bathymetric map.
- 6b – what is grey backdrop? Presumably topography? Needs stated though.
- Caption: "Glacier surface elevation catchments….."?
- Caption: "….elevation contours (grey)."
- 6a – what is the black line? The 0 m contour? Make this clear in caption.

**Figure 7**

- Needs a box showing its location on figure 6 (the extent of figure 6 is greater than that of figure 7).
- See colour scale comments for figure 6.

**Dr Neil Ross**
**Newcastle University**
**17th May 2018**

---

## Referee Comment (RC2) · Anonymous Referee #2 · 3 Jun 2018

The authors publish the data as a dataset in addition to another paper. Ice thickness grids is a fundamental parameter for many glaciological applications, and also useful for other purposes. My main objection to the paper in its current state is that I miss the point data of ice thickness that are very valuable for researchers and projects such as demonstrated in the ITMIX project (Farinotti et al. 2017). There is an available database for ice thickness data, GlaThiDa, with a recent call out for GlaThiDa 3.0 (WGMS on cryolist on 2018-02-21, instructions on the website: http://wgms.ch/glathida_cfd/) and I suggest the authors prepare their dataset accordingly, publish it along with the grids in this data paper and refer to GlaThiDa in their paper. http://www.gtn-g.ch/data_catalogue_glathida/ This would be a real enrichment of the dataset and make it much more useful for researchers.

[Figure]

Data availability: It is written in the paper that 'the compiled data sets of ground-based and airborne radar surveys are freely available at:doi:10.21334/npolar.2017.702ca4a7'. This doi was not working and I could not assess it.

Figures

Figure 1. Add glacier basins for the five glaciers (from the recent Svalbard inventory). Makes it easier to see what mapped within each basin. Probably it is enough to have coordinates on two axes, e.g. below and right.

Figure 4. Difficult to see the location of the profiles in fig 1. I could not see it. Could here add the line that was digitized from the two profiles. Letter a and b are not on figure. Add to figure or add lower and upper in the figure text instead.

Figure 6. State surface elevation or bed elevation. Profiles(points) could been added to figure to show data source better.

Data: The datasets contained negative values of ice thickness. Ice thickness cannot be negative. There is no mentioning of this in the paper. The datasets need to be filtered, reviewed and resubmitted. As stated, the original datasets could be formatted to GlaThiDa and added to this paper, which would make it much more useful for researchers.

Reference:

Farinotti, D., Brinkerhoff, D. J., Clarke, G. K. C., Fürst, J. J., Frey, H., Gantayat, P., Gillet-Chaulet, F., Girard, C., Huss, M., Leclercq, P. W., Linsbauer, A., Machguth, H., Martin, C., Maussion, F., Morlighem, M., Mosbeux, C., Pandit, A., Portmann, A., Rabatel, A., Ramsankaran, R., Reerink, T. J., Sanchez, O., Stentoft, P. A., Singh Kumari, S., van Pelt, W. J. J., Anderson, B., Benham, T., Binder, D., Dowdeswell, J. A., Fischer, A., Helfricht, K., Kutuzov, S., Lavrentiev, I., McNabb, R., Gudmundsson, G. H., Li, H., and Andreassen, L. M.: How accurate are estimates of glacier ice thickness? Results from

[Figure]

ITMIX, the Ice Thickness Models Intercomparison eXperiment, The Cryosphere, 11, 949-970, https://doi.org/10.5194/tc-11-949-2017, 2017.

---

## Author Comment (AC1) · 15 Jun 2018

**Response to referee #1 Neil Ross on "Subglacial topography, ice thickness, and bathymetry of Kongsfjorden, northwestern Svalbard"**

**K. Lindbäck et al.**

katrin.lindback@npolar.no

Dear Neil Ross,

On behalf of all the authors of this discussion paper, I would like to thank you for your time and detailed comments on our data. Your suggestions and comments have been acknowledged and our responses can be found below.

10    Kind regards,

Katrin Lindbäck

**RC1 General comments**

This is a very good manuscript that presents ice thickness and subglacial topography/bathymetry digital elevation models (DEMs) of a series of glaciers in NW Svalbard. The quality of the data is very good, and the methods of acquisition and
15    processing (and their description) are appropriate. Errors and uncertainties are outlined effectively. The DEMs are important for several purposes, the most important being for numerical modelling of glacier behaviour/evolution, with implications for global sea level rise.

**Author response**

We are grateful to Neil Ross for his positive comments.

20    ## RC1.1

I do have some suggestions for ways in which the manuscript can be improved before publication:

1.  Abstract and conclusions are very superficial and general at present. Both need improvement so that they actually report/summarise the manuscript and provide more detail. Abstract is currently very short, so there is room to develop it.

**Author response**

We have extended the abstract with additional information.

**RC1.2**

2. The figures: (a) have multiple basic errors (e.g. lack easting northing, have bizarre distance scales etc.); (b) could be improved with some simple changes to the GIS (e.g. have discrete/classified colour scales, rather than continuous ones), and (c) are lacking the presentation of certain datasets (e.g. ice surface elevation). Detailed suggestions of how to edit/improve the figures are provided below.

**Author response**

We have edited the figures, see specific responses below under the detailed comments.

**RC1.3**

3. Results section is currently perfunctory (7 lines), whilst significant parts of the discussion section simply describe the data rather than interpret or discuss it. I recommend that sections 3&4 are merged into a single section entitled something along the lines of "Description of DEM morphology and implications for future glacier and landscape dynamics in NW Svalbard"

**Author response**

We have merged section 3 and 4 into a single section titled "Results".

**RC1.4**

4. The authors should consider including some profiles extracted from the DEMs as figures. This would serve 2 purposes: (a) to illustrate to those not familiar with the study area and dataset the 'morphology' of the ice thickness/bed elevation; (b) to qualitatively demonstrate the quality of the data (e.g. are there any artefacts at critical locations?). I suggest that along ice flow profiles down the centre lines of the glaciers would be useful as (a) such profiles would likely be input data for 2D ice flow models; and (b) the discussion section describes sills and overdeepenings that are not necessarily obvious from the DEMs alone (at least to someone not familiar with the datasets).

**Author response**

We have included a figure ice surface and subglacial elevation along the flow lines of each glacier.

**RC2 Detailed Comments - Abstract**

**RC2.1**

P1. L12-13: "..which will affect fjord circulation and ecosystems…." – how will retreating glaciers do this? If this is the justification for the datasets, then you need to explain how. I'd also suggest adding something about global sea level, which is the most important impact of your dataset (i.e. DEM can be used for numerical modelling of future glacier behaviour, from which future sea level can be modelled), and perhaps about insights into surging glacier behaviour.

**Author response**

We have added a section in the introduction describing how retreating glaciers can affect fjord circulation and ecosystems.

**RC2.2**

P1. L13-15: it is worth inserting "ice-penetrating radar" into this sentence (i.e. after "ground-based").

**Author response**

We have rephrased the sentence as suggested.

**RC2.3**

P1. L15: One sentence on findings. This is not enough. I also suggest a re-phrasing to "Three of the glaciers would have to retreat by ~10 km…..". It would also be worth naming those three glaciers.

**Author response**

We have rephrased the sentence as suggested above and added the names of the glaciers.

**RC2.4**

P1. L16-17: "…will be valuable for future studies of glacier dynamics, geology, hydrology and fjord circulation". Fair enough, but how and why? The authors never discuss why the data would be valuable for many of these topics in the manuscript, so why make the statement here? Perhaps the discussion section of the manuscript could be expanded to develop this justification though?

**Author response**

We have mentioned a couple of studies where the dataset has been used for such purposes in the results section: "The high-resolution subglacial DEM of Kronebreen has so far been used in studies of basal sliding, subglacial hydrology and calving (How et al., 2017; Vallot et al., 2017, 2018)." This is a dataset paper and the aims and scope of the journal states (https://www.earth-system-science-data.net/about/aims_and_scope.html): "Articles in the data section may pertain to the planning, instrumentation, and execution of experiments or collection of data. Any interpretation of data is outside the scope of regular articles." Therefore, we leave such discussion to potential future papers.

**RC3 Detailed Comments – Introduction**

**RC3.1**

P1. L26-27: Perhaps add "..providing a contribution to rising global sea levels."?

**Author response**

We have rephrased the sentence as suggested.

**RC3.2**

P2. L1: The glaciers could advance too (particularly if they are of surge-type), so perhaps "glacier dynamics" or "glacier oscillations" rather than "glacier retreat"?

**Author response**

We have added "glacier dynamics" to the sentence.

**RC3.3**

Do any of the glaciers in the study area surge? It might be worth stating whether this is the case or not. I believe that Kongsvegen is surge-type glacier?

**Author response**

We have added a sentence describing observed surges.

**RC4 Detailed Comments – Study area**

**RC4.1**

P2. L9: "Maximum depth in the outer part of the fjord…."?

**Author response**

We have rephrased the sentence as suggested.

**RC4.2**

I'd would have liked to have seen more information on the glaciology of these glaciers (e.g. surging, subglacial sediments, thermal regime etc.), or at least more references to published papers that describe the glaciological characteristics of these in detail (e.g. I am aware of papers by John Woodward/Tavi Murray/Adam Booth on Kongsvegen), and perhaps at least some description of the wider controls on the glacial system in the study area (e.g. temperature/precipitation/ oceanography etc.).

**Author response**

As stated above under comment RC2.4 this is a dataset paper and we believe detailed information as described in the comment is out of the scope of this kind of paper.

**RC5.1 Detailed Comments – Data and methods**

**RC5.1.1**

P2. L18: No need for "have"
P2. L19-20: "…high a radar frequency…"?
P2. L25: "are" rather than "is"?

**Author response**

We have rephrased the sentence as suggested.

**RC5.1.4**

P2. L26: There are much better references than Bamber et al and Fretwell et al. I recommend that examples that report individual airborne surveys are referenced, rather than those that report Antarctic- and Greenland-wide DEMs.

**Author response**

There are many references within these papers to individual datasets, therefore we added "and references therein" to guide the readers to these papers.

**RC5.2 Detailed Comments – Data and methods: Radar data collected after 2014**

5 **RC5.2.1**

P3. L7: Can the authors provide more information on the transmitter? Later in the manuscript they refer to Kentech and Narod transmitters, but they do not describe this one. Is it a bespoke transmitter built by NPI? Please either state that it is a bespoke system, or, if it is an 'off-the-shelf' system, please give its name (e.g. Narod etc.).

**Author response**

10 We have added details about the transmitter.

**RC5.2.2**

P3. L9: Break this into two sentences: "…different sensitivity ranges. One channel was attenuated by……"

**Author response**

We have rephrased the sentence as suggested.

15 **RC5.2.3**

P3. L16: still 125 traces stacked (i.e. equivalent to acquisition with airborne system)?

**Author response**

Yes, we have adjusted the sentence accordingly.

**RC5.2.4**

20 P3. L19: 15 m in front or behind the midpoint?

**Author response**

In front, we have adjusted the sentence accordingly.

**RC5.2.5**

P3. L24: "…rubber-band correction to re-sample the data to a uniform…"?

**Author response**

We have rephrased the sentence as suggested.

5 **RC5.2.6**

P3. L26: "amplify" rather than "reinforce"?

**Author response**

We have rephrased the sentence as suggested.

**RC5.2.7**

10 P3. L28: The velocity of the radio wave through the ice is assumed (and assumes cold ice?). Can the authors justify this assumption in anyway? I note that Woodward et al., Journal of Glaciology, 2003 reports CMP measurements on Kongsvegen that could be referenced.

**Author response**

We have justified and included this in the error estimates. We have added the suggested reference above.

15 **RC5.3 Detailed Comments – Data and methods: Radar data collected prior to 2014:**

**RC5.3.1**

P4. L6: Delete "possibly"

**Author response**

We have rephrased the sentence as suggested.

20 **RC5.3.2**

P4. L14-15: It might have just been me, but I didn't really follow this sentence. Consider rewording.

**Author response**

We have rephrased the sentence.

**RC5.3.3**

P4. L23: how much stacking?

**Author response**

We unfortunately don't have any information on the post processing stacking number.

**RC5.4 Detailed Comments – Data and methods: Surface and bathymetric elevation data:**

**RC5.4.1**

P6. L9: "The offshore bathymetric…"?

**Author response**

We have rephrased the sentence as suggested.

**RC5.4.2**

P6. L9: "acquired" rather than "captured"?

**Author response**

We have rephrased the sentence as suggested.

**RC5.4.3**

P6. L9-11: are there any references for this dataset (e.g. a technical report). If there are not, then perhaps a more detailed description of acquisition and processing is required within this manuscript?

**Author response**

We unfortunately don't have any more information or technical report about the dataset.

**RC6 Detailed Comments – Results**

**RC6.1**

P7. L9: "For Kronebreen….."?

**Author response**

We have rephrased the sentence as suggested.

**RC6.2**

P7. L10: Are these 50 m resolution DEMs also available via the NPI data centre website?

**Author response**

Yes, it is available and clarified in the text under Conclusion/Summary (Section 6).

**RC7 Detailed Comments – Discussion**

**RC7.1**

P7. L12-13: Reword to: "…knowledge of the subglacial topography of retreating glaciers…"

**Author response**

We have restructured the Results and Discussion section into one, and therefore removed this sentence.

**RC7.1**

P7. L15-P8. L7: Lots of this material is description of the DEM morphology, so should be in a results section not a discussion section. In addition, there are multiple features (e.g. overdeepenings)/place names (e.g. Steindolpen) referred to that are not annotated on any figure. To improve readability, the authors should annotate appropriate figures.

**Author response**

We have merged the Discussion section into the Results section. We have added a 2D plot figure along the flow line, where overdeepenings are more visible. We have removed the place names from the text.

**RC7.2**

Much of the content describing the morphology of the DEMs would be helped by the inclusion of profiles (e.g. along ice flow, down the centre line of the glaciers) that would help to illustrate features such as overdeepenings and sills. You might also want to cite some classic literature in this section too (e.g. Holtedahl 1967 https://www.tandfonline.com/doi/abs/10.1080/04353676.1967.11879749)

**Author response**

We have added a 2D plot figure along the flowlines of the glaciers.

**RC7.2**

P7. L22: Perhaps include a satellite image in the manuscript, and annotate the rock outcrop? I would suggest that a satellite image of the entire study area would be useful (e.g. figure 1).

**Author response**

We have removed the sentence. We have not included a satellite image, since it is not one of the products of this paper, but satellite images are easily accessible at e.g. toposvalbard.npolar.no.

**RC7.3**

P7. L22-24: Might have been me, but this sentence didn't seem easy to disentangle. I suggest the authors consider rewording it.

**Author response**

We have removed the sentence.

**RC7 Detailed Comments – Conclusion**

Like the abstract, rather generic and superficial. I suggest that the authors expand the conclusion to report on/summarise the dataset and manuscript in more detail.

**Author response**

We have renamed the section to "Summary" and added some additional text.

**RC8 Detailed Comments – Author contributions**

P8. L21. "was the main responsible"? Do the authors mean "was primarily responsible"?

**Author response**

We have rephrased the sentence as suggested.

5 **RC9 Detailed Comments – Table 1**

Is it also worth reporting standard deviation?

**Author response**

We have added standard deviations.

**RC10.1 General Comments – Figures**

10 **RC10.1.1**

All maps need 'easting (utm)' and 'northing (utm)'. Currently only figure 5 has this.

**Author response**

We have added x and y labels to all the figures.

**RC10.1.2**

15 The distance scale on all of the maps should extend upwards from 0, rather than having ranges from negative values to positive values (e.g. fig 1 extends from -5 to 5 km)

**Author response**

We have changed the distance scales to run from 0.

20 **RC10.1.3**

"Grid projection is Universal Transverse Mercator Zone 33W"

**Author response**

We have rephrased the sentence as suggested.

**RC10.1.4**

Maps should be annotated with features/places referred to in the text (e.g. Steindolpen nunatak)

5 **Author response**

We have removed the place names in the text.

**RC10.2 Detailed Comments – Figure 1**

**RC10.2.1**

I could not make out the location of: (a) 2005 data; or (b) Black lines indicating location of profiles in figure 4.

10 **Author response**

The lines had gone missing in the figure, we have added them again.

**RC10.2.2**

A satellite image of the study area would be useful.

**Author response**

15 We have not included a satellite image, since it is not one of the data products, but can be easily accessed at toposvalbard.npolar.no.

**RC10.2.3**

It would be useful to show which survey lines were (a) acquired by helicopter; (b) acquired by skidoo.

**Author response**

20 We have not done this since it would add too much information to the already crowded figure. We have described in the text that the lines on Kongsvegen where acquired by snowmobile.

**RC10.2.4**

Rather than stating "blue areas are sea, green areas land and white glacierized", why not add these to the legend?

**Author response**

We have added the suggestion to the legend.

**RC10.3 Detailed Comments – Figure 3**

**RC10.3.1**

5  This (& fig 2) is a really useful figure. It should not be removed from the manuscript. Could figures 2 & 3 be integrated into one figure however?

**Author response**

We have merged the images into one as suggested.

**RC10.3.2**

10  There is a little ambiguity to me about the position of the antennas. Do they run from the Tx and Rx outwards to the end of the plastic pipes, or are they just contained within the plastic pipes. Can the authors please make this clear?

**Author response**

We have clarified the antenna locations in the text caption.

**RC10.3.3**

15  What about a scale bar?

**Author response**

We have added a scale bar.

**RC10.4 Detailed Comments – Figure 4**

**RC10.4.1**

20  Where are these radagrams located? I could not see them on figure 1.

**Author response**

We have added the locations of the radargrams in Figure 1 with black lines.

**RC10.4.1**

Perhaps the authors could annotate key features in the radargrams for non-experts in ice-penetrating radar (e.g. ice surface, ice-bed interface, internal layering, hyperbolic reflections from englacial conduits etc.). Perhaps also worth annotating surface multiples?

5  **Author response**

We have annotated the surface and bed reflector in the radargrams. We have chosen not to annotate other features, since we don't mention these in the text.

**RC10.4.1**

Figure 4a: Can the authors explain why if the data have been migrated (as stated in the methods section) there are still
10  hyperbolic reflections within the ice column? Is this evidence for warm ice with a different velocity to that used in the migration? It looks like there might be hyperbola at the bed too (i.e. between 0-2 km).

**Author response**

We believe this is caused by off-nadir reflections such as crevasses close to the front.

**RC10.5 Detailed Comments – Figure 5**

15  No units are given? I assume metres?

**Author response**

We have added units to the figure.

**RC10.6 Detailed Comments – Figure 6**

**RC10.6.1**

20  Needs a box showing its location on figure 1 (the extent of figure 1 is greater than that of figure 6).

**Author response**

We have added a box in Figure 1 to show the location of Figure 6.

**RC10.6.2**

Edit the colour scales so that they are in discrete intervals of equal spacing (e.g. 100 m intervals), so that the boundaries between the colours match the contours. At present, with a continuous colour scale it is very difficult for the reader to match the colours with actual elevations. The inclusion of some topographic/bathymetric profiles (see earlier comments) may help with this.

**Author response**

We have not edited the colour scales to discrete intervals, since this removes a lot of details in the to be more detailed. We have also included a figure with 2D plots of ice surface and subglacial elevation along the flowlines.

**RC10.6.2**

The authors report a surface DEM in the manuscript. This should be displayed in this figure. They may also wish to include an ice velocity map if one exists for this region.

**Author response**

The surface DEM is not one of the data products, therefore we don't want to have this in a figure. The same goes for a velocity map.

**RC10.6.2**

6a – it needs to be made clearer that this is the subglacial/bathymetric map.

**Author response**

We have clarified in the figure caption that the map shows subglacial, bathymetric and land elevation.

**RC10.6.2**

6b – what is grey backdrop? Presumably topography? Needs stated though.

**Author response**

The backdrop in the updated image is the hill shade from map 6a. We have clarified this in the figure caption.

**RC10.6.2**

Caption: "Glacier surface elevation catchments….."?

**Author response**

We have added the suggestion to the figure caption.

**RC10.6.2**

Caption: "….elevation contours (grey)."

5    **Author response**

We have adjusted the figure caption.

**RC10.6.2**

6a – what is the black line? The 0 m contour? Make this clear in caption.

**Author response**

10    We have edited the legend to make this clearer.

**RC10.7 Detailed Comments – Figure 7**

**RC10.7.1**

Needs a box showing its location on figure 6 (the extent of figure 6 is greater than that of figure 7).

**Author response**

15    We have added a box in Figure 1 to show the extent of figure 7.

**RC10.7.2**

See colour scale comments for figure 6.

**Author response**

We have edited the color scales.

20

**Response to referee #2 on "Subglacial topography, ice thickness, and bathymetry of Kongsfjorden, northwestern Svalbard"**

**K. Lindbäck et al.**

**katrin.lindback@npolar.no**

Dear Referee,

On behalf of all the authors of this discussion paper, I would like to thank you for your comments on our data. Your suggestions and comments have been acknowledged and our responses can be found below.

10 Kind regards,

Katrin Lindbäck

**RC1 General comments**

The authors publish the data as a dataset in addition to another paper. Ice thickness grids is a fundamental parameter for many glaciological applications, and also useful for other purposes. My main objection to the paper in its current state is that I miss

15 the point data of ice thickness that are very valuable for researchers and projects such as demonstrated in the ITMIX project (Farinotti et al. 2017). There is an available database for ice thickness data, GlaThiDa, with a recent call out for GlaThiDa 3.0 (WGMS on cryolist on 2018-02-21, instructions on the website: http://wgms.ch/glathida_cfd/) and I suggest the authors prepare their dataset accordingly, publish it along with the grids in this data paper and refer to GlaThiDa in their paper. http://www.gtn-g.ch/data_catalogue_glathida/ This would be a real enrichment of the dataset and make it much more useful

20 for researchers.

**Author response**

We are grateful to the referee for the comments. We will happily share the dataset to the Glacier Thickness Database, as soon as our dataset has been published, and will prepare the data accordingly. Then, hopefully, the GlaThiDa dataset can refer to our publication for detailed methods and error analysis. Our dataset not only includes ice thickness data, but also subglacial

25 and bathymetric data, which are valuable for a wide range of studies. The dataset will be updated when the quality of the data is improved or if new data sets become available.

**RC2 Detailed Comments – Data availability**

It is written in the paper that 'the compiled data sets of ground-based and airborne radar surveys are freely available at: doi:10.21334/npolar.2017.702ca4a7'. This doi was not working and I could not assess it.

**Author response**

5   The link was working when we tested it. The data set is freely available at Norwegian Polar Data Centre (doi: 10.21334/npolar.2017.702ca4a7).

**RC3 Detailed Comments – Figures**

**RC3.1**

Figure 1. Add glacier basins for the five glaciers (from the recent Svalbard inventory). Makes it easier to see what mapped
10   within each basin. Probably it is enough to have coordinates on two axes, e.g. below and right.

**Author response**

We show glacier basins in Figure 6b. Figure 1 is already crowded and we would not like to add more information to this figure.

**RC3.2**

Figure 4. Difficult to see the location of the profiles in fig 1. I could not see it. Could here add the line that was digitized from
15   the two profiles. Letter a and b are not on figure. Add to figure or add lower and upper in the figure text instead.

**Author response**

Thanks for noticing. We have corrected Figure 1 with the locations of the profiles in Figure 4 (black lines).

**RC3.2**

Figure 6. State surface elevation or bed elevation. Profiles (points) could been added to figure to show data source better.

20   **Author response**

We have updated the figure caption to clarify that the elevation refers to subglacial, bathymetric and land elevation. We prefer to show the gridded DEM since that is the data product.

**RC4 Detailed Comments – Data**

Data: The datasets contained negative values of ice thickness. Ice thickness cannot be negative. There is no mentioning of this in the paper. The datasets need to be filtered, reviewed and resubmitted. As stated, the original datasets could be formatted to GlaThiDa and added to this paper, which would make it much more useful for researchers.

5 **Author response**

We have filtered the data so that there are no outliers and ice thickness starts at 0. As stated above we will happily share the dataset with GlaThiDa after publication.